# Effects of periconceptional ethanol on mitochondrial content and oxidative stress in maternal liver and placentas from male and female fetuses in rats

Sarah S. Steane ⬥, Tulika Das, Jacinta I. Kalisch-Smith ⬥, Dinithi T. Mahaliyanage, Lisa K. Akison ⬥, Karen M. Moritz ⬥ and James S. M. Cuffe ⬥

*School of Biomedical Sciences, The University of Queensland, Brisbane, QLD, Australia*

Handling Editors: Laura Bennet & Janna Morrison

The peer review history is available in the Supporting Information section of this article (https://doi.org/10.1113/JP287566#support-information-section).

**Abstract figure legend** Alcohol consumption around conception (Periconceptional ethanol exposure or PC:EtOH) is highly prevalent with many pregnant people stopping or reducing alcohol consumption once they realise they are pregnant. A rat model of PC:EtOH is shown in which rats are provided 12.5% ethanol from 4 days prior to conception on embryonic day (E) –4 until 4 days after conception on E4. PC:EtOH reduces mitochondrial content and expression of alcohol metabolising genes catalase (*Cat*) and cytochrome P450 2E1 (*Cyp2e1*) in the maternal liver at E5 without impacting markers of oxidative stress. Mitochondrial content remained low in the maternal liver at E20. PC:EtOH induced a subtle reduction in mitochondrial content and increased superoxide dismutase (SOD) activity in the junctional zone of placentas. It remains unknown how such changes to mitochondrial number in the maternal liver impacts maternal physiology after pregnancy. By contrast, mitochondrial deficits in the placenta may contribute to the adult onset cardiometabolic disease previously demonstrated in this model of PC:EtOH. Created with https://BioRender.com.

S. Steane and T. Das contributed equally to this work.

The Journal of Physiology

**Abstract** Alcohol exposure during pregnancy disrupts fetal development and programs life-long disease. We have shown, in rats, that alcohol exposure during the periconceptional period (PC:EtOH), causes placental dysfunction and cardiometabolic disease in offspring. The process of metabolising alcohol can cause oxidative stress and damage mitochondrial DNA (mtDNA). It is unknown whether alcohol metabolism in a rat model of PC:EtOH impacts oxidative stress markers and mitochondrial content in maternal and placental tissues. We aimed to determine whether PC:EtOH induced oxidative stress and reduced mtDNA in maternal liver and the placental labyrinth and junctional zone. Sprague–Dawley rats were given an ethanol liquid (12.5% v/v) or control (0%) diet for one oestrous cycle before mating to embryonic day (E) 4. Maternal livers were collected at E5 and E20. Placentas were collected at E20 and separated into the junctional zone and labyrinth zone. PC:EtOH reduced *Cyp2e1* mRNA levels and mtDNA in the E5 liver with lower mtDNA persisting to E20, at which time mitochondrial proteins were also decreased. PC:EtOH also reduced mitochondrial content in the E20 junctional zone, although mitochondrial protein levels were unaffected. Superoxide dismutase activity was increased in the placental junctional zone and there was no evidence of oxidative stress. The present study demonstrates that alcohol exposure around conception, reduces mitochondrial content within the maternal liver and the junctional zone of the placenta towards the end of pregnancy. These prolonged deficits may have disrupted metabolic processes required for a healthy pregnancy. The study further supports avoiding alcohol when planning a pregnancy.

(Received 28 August 2024; accepted after revision 23 January 2025; first published online 10 February 2025)

**Corresponding author** J. S. M. Cuffe: Sir William MacGregor Building, The University of Queensland, Brisbane, QLD, 4072, Australia. Email: j.cuffe1@uq.edu.au

## Key points

- Even when alcohol is consumed only around conception (PC:EtOH), it can have profound impacts on the developing baby.
- Here, we use our established rat model to investigate if PC:EtOH causes oxidative stress and reduces mitochondrial content in the maternal liver immediately after exposure on embryonic day (E) 5. We also investigate these parameters at the end of pregnancy (E20) in maternal liver and the placenta.
- PC:EtOH reduced mitochondrial DNA content in the maternal liver by 77% at E5 and by 40% at E20. At E20, expression of proteins that form the electron transport chain were also reduced. The placenta had a more subtle reduction in mitochondrial DNA content, but protein levels of mitochondrial complexes were unchanged.
- There was no evidence of oxidative stress in the maternal liver or placenta in response to PC:EtOH. The lack of oxidative stress in the placenta may be a result of compensatory increases in anti-oxidants.

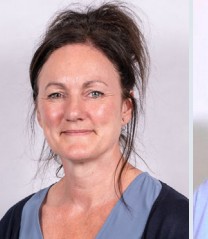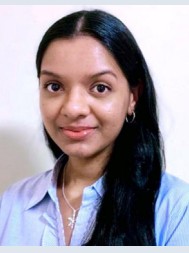

**Sarah Steane** recently completed a PhD at the University of Queensland investigating the effects of alcohol exposure in early pregnancy on placental and fetal outcomes in both a rodent model and in a clinical cohort. Her thesis demonstrated that alcohol exposure around conception was associated with alterations in placental levels of methyl donors, DNA methylation and gene expression. These findings suggest that early pregnancy alcohol exposure disrupts placental development, which probably contributes to negative impacts on the fetus. **Tulika Das** is a recent Masters of Biotechnology graduate from The University of Queensland. She previously also completed an Honours Degree in Biomedical Science supervised by Dr James Cuffe, Dr Lisa Akison and Professor Karen Moritz in the Developmental Programming in Disease Laboratory in the School of Biomedical Sciences. Her research focused on understanding the effect of prenatal alcohol exposure on the development of long-term offspring disease, with a specific interest in the programming of oxidative stress. She is currently in the biopharmaceutical industry and has an active interest in improving health outcomes and delivering quality therapies to help shape the health of future generations.

# Introduction

Approximately 50% of pregnant people in Australia consume ethanol during early gestation or before pregnancy recognition, with binge and heavy alcohol use during this time being more common than low or moderate consumption (Callaghan et al., 2003; McCormack et al., 2017). It is well established that ethanol exposure during pregnancy can lead to a range of poor outcomes for children, ranging from the severe neurological deficits seen in individuals with fetal alcohol syndrome (FAS) through to the wider spectrum of deficits across all body systems seen in individuals with fetal alcohol spectrum disorder (FASD). There are probably also many ethanol exposed individuals with subtle changes to physiology that do not fit into any formal diagnosis of FASD. Severity of outcomes has been related to the timing and amount of ethanol exposure, as well as maternal micronutrient status (Petrelli et al., 2018; Steane et al., 2023).

Ethanol readily passes through the plasma membrane surrounding cells and so, when consumed during pregnancy, maternal tissues, developing placenta and the growing embryo/fetus are all exposed (Heller & Burd, 2014). Alcohol is even able to pass into oviductal and uterine secretions (Sandor et al., 1981), which suggests that alcohol exposure during early pregnancy will result in the embryo being bathed in secretions that contain alcohol. Using our well-established rat model of peri-conceptional ethanol exposure (PC:EtOH), we have demonstrated that ethanol exposure specifically around conception impacts the embryo, altering blastocyst cell pluripotency and markers of DNA methylation (Kalisch-Smith et al., 2017, 2019). We have also shown that ethanol during this time induces structural changes in the placenta that are present towards the end of pregnancy (Gardebjer et al., 2014; Kalisch-Smith et al., 2017) even though ethanol exposure ceased before formation of the placenta. Offspring from this model develop a range of metabolic (Gardebjer et al., 2018), cardiovascular (Dorey et al., 2023) and neurological deficits (Burgess et al., 2019, 2020) in adult life. These long-term changes to placental and fetal development are probably caused by a combination of the direct effects of ethanol on the early embryo and changes in maternal physiology that persist beyond the period of ethanol exposure.

A key pathway through which alcohol may cause tissue damage is via its capacity to induce oxidative stress. Ethanol induces oxidative stress via multiple processes including changes to the $NAD^+/NADH$ ratio as a result of ethanol metabolism or mitochondrial damage caused by either ethanol or its metabolite acetaldehyde (Bailey & Cunningham, 2002; Wu & Cederbaum, 2003). The main site of ethanol metabolism during pregnancy is in the maternal liver, with the primary pathway initiated by oxidation of ethanol to acetaldehyde by alcohol dehydrogenase (ADH). This process requires $NAD^+$ as the oxidising agent, generating NADH. A secondary pathway involves oxidation by cytochrome P450 enzymes, notably CYP2E1, which generates a large amount of oxygen radicals (Lu & Cederbaum, 2008). A third more minor pathway requires oxidation of ethanol in peroxisomes via hydrogen peroxide ($H_2O_2$). The acetaldehyde produced by these mechanisms is highly reactive and capable of causing tissue damage (Ciafre et al., 2020). It is metabolised to acetate by acetaldehyde dehydrogenase (Aldh1a1 and Aldh2) generating NADH in the process. These processes largely occur within the liver and the generated reactive oxygen species (ROS) can cause oxidative stress, damage to mitochondrial DNA (mtDNA) and ultimately decreases in mitochondrial protein levels (Hoek et al., 2002). This can impact the capacity of the liver to utilise energy substrates appropriately and may contribute to a range of liver related conditions. When ethanol is consumed during early pregnancy, its metabolism may impair liver function and lead to liver related problems during pregnancy (Lee et al., 2020).

Ethanol is metabolised to a lesser extent by the placenta (Sanchis & Guerri, 1986). When ethanol is consumed during late pregnancy, oxidative free radicals formed during ethanol metabolism may contribute to placental damage (Martin-Estal et al., 2022). However, when ethanol exposure occurs early in pregnancy, ethanol probably directly induces mitochondrial damage in the developing blastocyst and/or embryo. Studies have shown that acetaldehyde can accumulate in uterine tissue following alcohol exposure and so it is also possible that the embryo may be exposed to high levels of acetaldehyde during early alcohol exposure (Buthet et al., 2011). Mitochondrial biogenesis is known to be maximal around the time of implantation where mitochondria within the trophectoderm are being rapidly produced (May-Panloup et al., 2021). Disruption to mitochondria at this time may have long-term consequences for the developing embryo and contribute to the altered placental biology later in gestation (Gardebjer et al., 2014). Dysfunctional mitochondria generate excess oxygen free radicals, and endogenous antioxidant systems such as superoxide dismutase (SOD) within the placenta would need to adapt to prevent oxidative stress (Lu & Sferruzzi-Perri, 2021).

The present study used a rodent model of peri-conceptional ethanol exposure to examine the expression of ethanol metabolising genes and mitochondrial content in the maternal liver both immediately following exposure and towards the end of pregnancy. We also characterised how periconceptional ethanol exposure impacts mitochondrial content in the placenta close to term and whether antioxidant systems adapt to prevent oxidative stress in the placenta.

## Methods

### Animal treatment and tissue collection

All animal experimentation was approved by the University of Queensland Animal Ethics Committee in accordance with the Australian Code for the Care and Use of Animals for Scientific Purposes (approval numbers: AE000465/467) (National Health & Medical Research Council, 2018). All animal treatment and tissue collection protocols have been previously described (Gardebjer et al., 2014; Kalisch-Smith et al., 2017) and are graphically presented in Fig. 1. Briefly, female Sprague–Dawley rats were monitored daily to determine stage of the oestrous cycle. Rats were placed on either a liquid diet containing 12.5% v/v ethanol (PC:EtOH, $n = 17$) or containing 0% EtOH (Control, $n = 16$) for one oestrous cycle prior to mating. The diet was freshly prepared daily and provided for 21 h day$^{-1}$, after which the diet was removed and replaced with standard drinking water for the remaining 3 h day$^{-1}$. We previously showed that animals given this diet reach a peak in blood alcohol content of ~0.18%, 30 min after being offered the diet, although this quickly drops down to 0.07% and 0.05% at 3 and 5 h, respectively (Gardebjer et al., 2014). Four days later, at proestrous, rats were mated and the presence of a seminal plug the following morning confirmed successful pregnancy. This time point was designated to be embryonic day (E) 0.5. The liquid diets continued until the end of E4 when both groups were placed back on a standard chow diet (Rat &

Mouse Meat-Free Diet; Specialty Feeds, Memphis, TN, USA) until either E5 (blastocyst stage) when animals were humanely killed via decapitation ($n = 9$ control and 8 PC:EtOH) or E20 when animals were humanely killed with a lethal dose of ketamine/xylazine ($n = 7$ control and 9 PC:EtOH). Maternal liver samples were collected at E5 and E20 and placental tissue collected at E20. Placentas were divided into the junctional zone (JZ) and labyrinth zone (LZ), and all tissues were snap frozen. Fetal sex was confirmed using quantitative PCR (qPCR) as previously described (Bartho et al., 2019). One male and one female placenta was randomly selected from each litter and used for subsequent analyses.

### Analysis of gene expression and estimation of mtDNA content

Methods used for analysis of gene expression have been previously described (Bartho et al., 2019). Briefly, RNA was extracted from liver and placental samples using RNeasy mini-kits (Qiagen, Hilden, Germany) in accordance with the manufacturer's instructions. cDNA was synthesised from 100 ng μL$^{-1}$ RNA using iScript Reverse Transcription Supermix (Bio-Rad, Hercules, CA, USA) and a C-1000 Thermal Cycler (Bio-Rad). qPCR was performed using Kiqstart SYBR Green primers (Sigma-Aldrich, St Louis, MO, USA) (Table 1) or TaqMan Real-Time PCR Assays (Adh1: Rn00570670_m1, Aldh1a1:Rn00755484_m1, Aldh2: Rn00583474_m1,

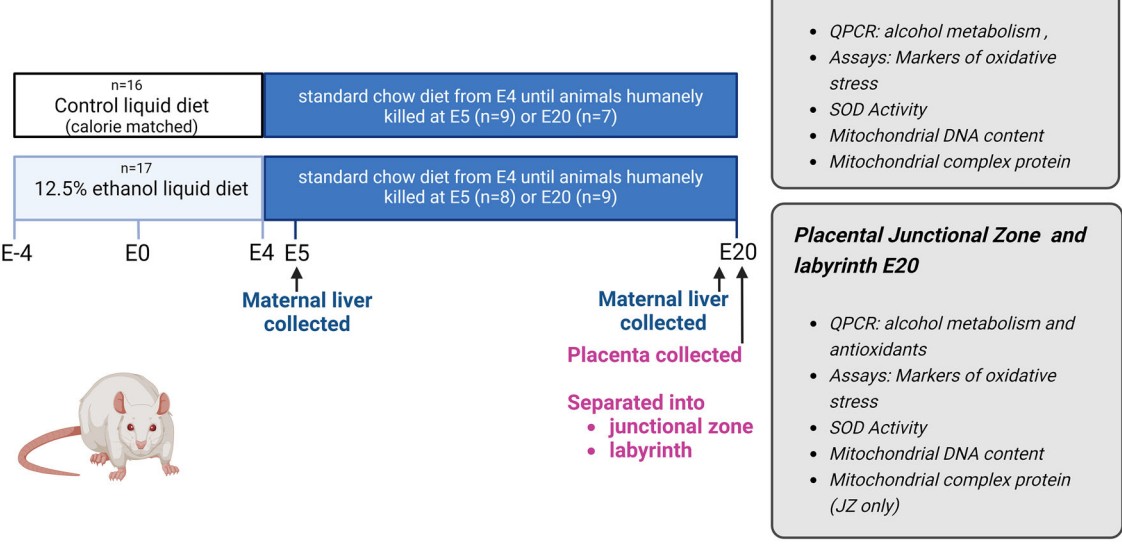

**Figure 1. Schematic demonstrating the experimental overview of our rat model of periconceptional ethanol exposure**
Female Sprague–Dawley rats were fed a liquid diet for 8 days around conception and were humanely killed either at embryonic day (E) 5 or at E20. Liver tissue was collected from the mother and placental tissue separated into JZ and LZ for analysis. Created with https://BioRender.com. [Colour figure can be viewed at wileyonlinelibrary.com]

**Table 1. Primer sequences (forward and reverse) for genes of interest and endogenous control genes in rat**

| Gene name | Oligo name | Sequence (5′- to 3′) | RefSeq ID |
|---|---|---|---|
| Superoxide dismutase 1 | FR1_Sod1 | AATGTGTCCATTGAAGATCG | NM_017050 |
| | RR1_Sod1 | CACATAGGGAATGTTTATTGGG | |
| Superoxide dismutase 2 | FR1_Sod2 | CTACGTGAACAATCTGAACG | NM_017051 |
| | RR1_Sod2 | TCCAGAAAATGCTGTGATTG | |
| Catalase | FR1_Cat | CAAGTTCCATTACAAGACTGAC | NM_012520 |
| | RR1_Cat | TTAAATGGGAAGGTTTCTGC | |
| Thioredoxin 1 | FR1_Txn1 | CTTCTTTCATTCCCTXTGTG | NM_053800 |
| | RR1_Txn1 | TTTATAGAACTGGAAGGTTCGC | |
| Beta-actin* | FR1_Actb | AAGACCTCTATGCCAACAC | NM_031144 |
| | RR1_Actb | TGATCTTCATGGTGCTAGG | |
| Succinate dehydrogenase complex flavoprotein subunit A* | FR1_Sdha | ACTATTATTGCTACTGGGGG | NM_130428 |
| | RR1_Sdha | CTGAACAAATTCTAAGTCCTGG | |

All primers are KiqStart primers sourced from Sigma-Aldrich. *Endogenous controls.

**Table 2. Primer sequences (forward and reverse) for markers of mitochondrial DNA (mtDNA) and nuclear DNA (nDNA) in rat**

| Marker name | mtDNA oligo markers | Sequence (5′- to 3′) | RefSeq ID |
|---|---|---|---|
| Mitochondrial displacement loop | FR1_Mitochondrial D-loop | GGTTCTTACTTCAGGGCCATCA | NC_001665 |
| | RR1_Mitochondrial D-loop | GATTAGACCCGTTACCATCGAGAT | |
| Mitochondrially encoded NADH:ubiquinone oxidoreductase core subunit 4 | FR1_mt-Nd4 | ATAATTATAACTAGCTCAATCTGC | NC_005089 |
| | RR1_mt-Nd4 | TCGTAGTTGGTGTTTGCTAG | |
| Mitochondrial DNA 1 | FR1_mtRnr2 | TTGATCAACGGACCAAGTTAC | NC_001665 |
| | RR1_ mtRnr2 | CTGGATTGCTCCGGTCTGA | |

| Marker name | nDNA oligo markers | Sequence (5′-3′) | RefSeq ID |
|---|---|---|---|
| Beta actin | FR1_BACTIN | GGGATGTTTGCTCCAACCAA | NM_031144 |
| | RR1_BACTIN | GCGCTTTTGACTCAAGGATTTAA | |
| Tet methylcytosine dioxygenase 3 | FR1_Tet3 | AAAATGAGAAATGTTGGTGC | NC_0 05103 |
| | RR1_Tet3 | AAGACATCTGAGAACAAACC | |

Cyp2e1: Rn00580624_m1) in a total volume of 10 μL containing SYBR Green/probe Quanti-Nova PCR Master Mix (Qiagen) and 20 ng of cDNA per reaction. The present study examined the expression of alcohol metabolising genes (*Adh1*, *Cyp2e1*, *Cat* and *Aldh1a1*) in the maternal liver, as well as alcohol metabolising (*Adh1*, *Aldh2* and *Cat*) and antioxidant genes (*Sod1*, *Sod2* and *Txn1*), in the placental LZ and JZ zone. qPCR data were normalised to the geometric mean of Actb and Sdha ($n = 7$–9 per treatment for maternal liver and $n = 6$–9 per treatment for placental tissues).

DNA was extracted from maternal livers and placenta tissues to estimate mitochondrial content in terms of the ratio of mtDNA:nDNA as previously described (Bartho

et al., 2019). Total DNA was extracted and qPCR used to quantitate three separate mtDNA regions (primers designed against *mt-Dloop*, *mt-Nd4* and *mt-Rnr2*). The geometric mean of the $C_t$ value for three amplified products was normalised to the geometric mean of the $C_t$ values of two nuclear DNA (nDNA) markers (*Actb* and *Tet3*) to calculate the ratio of mtDNA:nDNA using the $\Delta\Delta C_t$ method. For primer details, see Table 2. The results for experimental groups were normalised to the average of control group for liver samples and to the male control group for placental samples ($n = 7$–9 per treatment for maternal liver and $n = 4$–9 per treatment for placental tissues).

## Measurement of protein levels of subunits of mitochondrial complexes

Protein levels of key proteins that form mitochondrial complexes I–V were measured using western blotting in maternal liver at both E5 and E20, as well as in placental JZ tissue as previously described (Bartho et al., 2019). Briefly, protein samples (3 mg mL$^{-1}$) were electrophoresed on 12% SDS-PAGE gel ($n = 6$ per group). Purified rat heart mitochondria protein was used as a positive loading control. Protein samples were transferred to a poly(vinylidene difluoride) membrane, blocked in Li-Cor Odyssey (phosphate-buffered saline) blocking buffer and incubated overnight with the polyclonal total OXPHOS rodent antibody cocktail (dilution 1:1000; ab110413; Abcam, Cambridge, UK) that contains five separate antibodies designed for key subunits of each of the five mitochondrial complexes. The IRDye® donkey anti-mouse secondary antibody was applied the following day for 1 h (dilution 1:15,000; #925-32212; Li-Cor Biosciences, Lincoln, NE, USA). Membranes were then incubated with mAb rabbit beta-actin for 4–12 h (dilution 1:1000; #4970S; Cell Signaling Technology, Danvers, MA, USA). The secondary fluorescent antibody IRDye® goat anti-rabbit immunoglobulin G was applied and incubated for 1 h (dilution 1:20,000; #925-68021; Li-Cor Biosciences). Protein bands of interest were visualised using the Odyssey CLx Imaging System (Li-Cor Biosciences). Densitometry of proteins of interest was performed using the Image Studio Lite v5.2 Software (Li-Cor Biosciences) and was normalised to ACTB. Protein expression of complex IV was not reproducible using the liver samples available and so this data was not included for the liver analsis ($n = 6$ per treatment for maternal liver). All complexes were able to measured in the placental samples ($n = 6$–7 per treatment group).

## Measurement of markers of oxidative stress and antioxidant activity

Malondialdehyde (MDA) concentrations were assessed in maternal liver at E5 and E20 using a standard thiobarbituric acid reactive substances (TBARS) assay (700870; Cayman Chemical, Ann Arbor, MI, USA). Briefly, 20 mg of tissue was homogenised in 200 μL of RIPA buffer before 100 μL of sample or MDA standard was incubated with thiobarbituric acid diluted in acetic acid and sodium hydroxide. The reaction was initiated by heating the samples and the MDA concentrations determined by measuring absorbance at 530 nm ($n = 7$–8 per treatment for maternal liver).

The capacity for the maternal liver tissue at E5 and placental JZ tissue to generate $H_2O_2$ when SOD was added to the reaction was measured against a standard curve ranging from 0–30 μM prepared using 30% $H_2O_2$ (Merck Millipore, Burlington, MA, USA) and mitochondrial respiration media number 05 (MiR05) buffer as previously described (Bartho et al., 2019). The absolute concentration of $H_2O_2$ generated each time an assay is performed depends on a range of external factors and so cannot be compared between plates ($n = 7$–9 per treatment for maternal liver and $n = 6$–10 per treatment for placental tissues).

SOD activity was measured in the maternal liver at E5 and E20 and within the JZ of the placenta using a standard SOD activity kit (Cayman Chemical) in accordance with the manufacturers direction and as previously described (Fisher et al., 2021). Protein was extracted using a HEPES sucrose buffer and normalised to 3 mg mL$^{-1}$ protein. Conversion of the added tetrazolium salt to formazan dye in the presence of xanthine and xanthine oxidase was indicative of SOD activity when plotted against a SOD standard curve. Samples were also run in parallel with xanthine oxidase not added to the plate to establish background absorbance. Absorbance was measured at 440 nm and SOD activity was calculated ($n = 7$–8 per treatment for maternal liver and $n = 6$–9 per treatment for placental JZ).

Protein carbonyls were measured in maternal liver tissue using a commercially available ELISA kit in accordance with the manufacturer's instructions (10005020; Cayman Chemical; $n = 8$ per treatment). Relative protein carbonyl concentrations were assessed in placental LZ ($n = 6$–8 per group) and JZ samples ($n = 6$–9) as previously described (Bartho et al., 2019). Protein samples were incubated with 2,4-dinitrophenyl hydrazine (DNPH) and protein-bound DNP detected using a rabbit anti-DNPH antibody (catalog. no. D9656; Sigma-Aldrich). An anti-rabbit horseradish perxidase-linked antibody was applied, colour developed and absorbance measured at 450 nm.

## Statistical analysis

All statistical analyses of experimental data were conducted using Prism, version 7.0 (GraphPad Software Inc., San Diego, CA, USA). Data sets were checked for a normal distribution using the D'Agostino and Pearson test or Shapiro–Wilk test (for small sample size) before conducting parametric or non-parametric analysis as appropriate. For maternal data and placental mitochondrial protein content, a two-tailed unpaired Student's $t$ test (using Welch's correction as required for unequal variances) was conducted to assess statistical significance for each variable (Mann–Whitney test for non-parametric data), with Bonferroni's adjustment for multiple testing. Two-way ANOVA (with treatment and sex as factors) was used to analyse placental data (Kruskal–Wallis test for non-parametric data). If an effect

of treatment ($P_{trt} < 0.05$), sex ($P_{sex} < 0.05$) or an interaction between the two ($P_{trt} \times P_{sex} < 0.05$) was detected, then a Sidak multiple comparisons test was performed. Data are presented as the mean $\pm$ SD. $P < 0.05$ was considered statistically significant.

## Results

### Effect of PC:EtOH on alcohol metabolising genes, markers of oxidative stress and total SOD activity in the maternal liver

PC:EtOH exposure had minimal impact on the expression of ethanol metabolising genes within the liver at E5 or E20 (Fig. 2A) compared to controls. At E5, PC:EtOH reduced expression of *Cyp2e1* ($P = 0.014$) and *Cat* ($P = 0.04$) but had no effect on expression of *Adh1* ($P = 0.079$) or *Aldh1a1* ($P = 0.153$). At E20, PC:EtOH had no significant effect on expression of *Adh1* ($P = 0.343$), *Cyp2e1* ($P = 0.689$), *Cat* ($P = 0.516$) or *Aldh1a1* ($P = 0.826$) (Fig. 2B). PC:EtOH had no impact on MDA concentrations within the maternal liver at either E5 ($P = 0.335$) (Fig. 2C) or E20 ($P = 0.679$) (Fig. 2D). The capacity of the E5 liver homogenate to generate $H_2O_2$ following the addition of SOD was not affected by PC:EtOH treatment (Control = 13 $\pm$ 5.95, PC:EtOH = 11.22 $\pm$ 2.45, $P = 0.418$). SOD activity was not affected by PC:EtOH treatment when measured at E5 ($P = 0.966$) (Fig. 2E) or E20 ($P = 0.493$) (Fig. 2F). PC:EtOH had no effect on protein carbonyl concentrations at E5 ($P = 0.062$) (Fig. 2G) or E20 ($P = 0.150$) (Fig. 2H).

### Effect of PC:EtOH on mtDNA content and mitochondrial complex proteins in the maternal liver

PC:EtOH reduced the mtDNA content of the maternal liver by ~77% compared to controls ($P < 0.001$) (Fig. 3A) at E5 and mtDNA content remained 40% lower at E20 ($P = 0.011$) (Fig. 3B) despite alcohol cessation 15 days prior. Given mtDNA content was lower in the livers at E5, expression of proteins that contribute to key complexes of the mitochondrial electron transport chain were assessed. NDUFB8 (Complex I) protein expression was reduced by PC:EtOH compared to controls in E5 maternal livers ($P < 0.001$) (Fig. 3C), whereas SDHB (Complex II) was increased ($P = 0.003$) (Fig. 3C). PC:EtOH did not affect the protein expression of UQCRC2 (Complex III, $P = 0.548$) (Fig. 3C) or ATP5A (Complex V, $P = 0.458$) (Fig. 3C) in E5 maternal livers. Mitochondrial protein levels at E20 reflected the reduction in mtDNA content with PC:EtOH reducing protein levels of NDUFB8 (Complex I, $P = 0.008$) (Fig. 3D), SDHB (Complex II, $P = 0.007$) (Fig. 3D), UQCRC2 (Complex III, $P = 0.002$) (Fig. 3D) and ATP5A (Complex V, $P = 0.001$) (Fig. 3D) compared to controls.

### Effect of PC:EtOH on alcohol metabolising genes, mitochondrial content and antioxidants in the placental LZ

PC:EtOH did not alter mRNA expression of *Adh1* ($P_{trt} = 0.204$) (Fig. 4A) or *Aldh2* ($P_{trt} = 0.121$) (Fig. 4B) in the placental LZ at E20 although *Adh1* mRNA levels were higher in males than females ($P_{sex} = 0.017$). PC:EtOH did not affect the mtDNA content of the placental labyrinth compared to controls ($P_{trt} = 0.715$) (Fig. 4C). The sex of the fetus had no impact on mtDNA content of the placental LZ ($P_{sex} = 0.601$). PC:EtOH did not alter the mRNA expression of *Txn1* ($P_{trt} = 0.431$) (Fig. 4D), *Cat* ($P_{trt} = 0.618$) (Fig. 4E), *Sod1* ($P_{trt} = 0.849$) (Fig. 4F) or *Sod2* ($P_{trt} = 0.612$) (Fig. 4G). The sex of the fetus did not impact mRNA levels of *Txn1* ($P_{sex} = 0.910$), *Cat* ($P_{sex} = 0.418$), *Sod1* ($P_{sex} = 0.830$) or *Sod2* ($P_{sex} = 0.091$) within the labyrinth. Relative protein carbonyl levels (Fig. 4H) were not impacted by PC:EtOH exposure ($P_{trt} = 0.855$) or fetal sex ($P_{sex} = 0.519$).

### Effect of PC:EtOH on alcohol metabolising genes and antioxidants in the placental JZ

*Adh1* was not expressed in the JZ (cycle range 36 – undetected). PC:EtOH did not alter mRNA expression of *Aldh2* ($P_{trt} = 0.217$) (Fig. 5A) or *Cat* ($P_{trt} = 0.056$) (Fig. 5B) within the placental JZ compared to controls. PC:EtOH increased the mRNA expression of *Txn1* ($P_{trt} = 0.004$) (Fig. 5C) within the JZ compared to controls. Similarly, PC:EtOH increased mRNA levels of *Sod1* ($P_{trt} = 0.003$) (Fig. 5D) with *post hoc* analysis demonstrating that PC:EtOH increased Sod1 mRNA levels in the JZ of females ($P = 0.047$). PC:EtOH also increased mRNA expression of *Sod2* ($P_{trt} = 0.043$) (Fig. 5E) and enzymatic activity of SOD protein ($P = 0.037$) (Fig. 5H) in the JZ of the placenta at E20 compared to controls. Fetal sex did not impact the mRNA levels of any genes or total SOD activity. Relative protein carbonyl levels ($P_{trt} = 0.184$, $P_{sex} = 0.571$) (Fig. 5G) and $H_2O_2$ concentrations ($P_{trt} = 0.237$, $P_{sex} = 0.588$) (Fig. 5H) were not impacted by PC:EtOH exposure or fetal sex.

### Effect of PC:EtOH on mtDNA content and mitochondrial proteins in the placental JZ

PC:EtOH reduced mtDNA content within the placental JZ ($P = 0.011$) (Fig. 6A). *Post hoc* analysis demonstrated that PC:EtOH reduced mtDNA content in the placentas of female fetuses by ~18% ($P = 0.016$). Given the limited

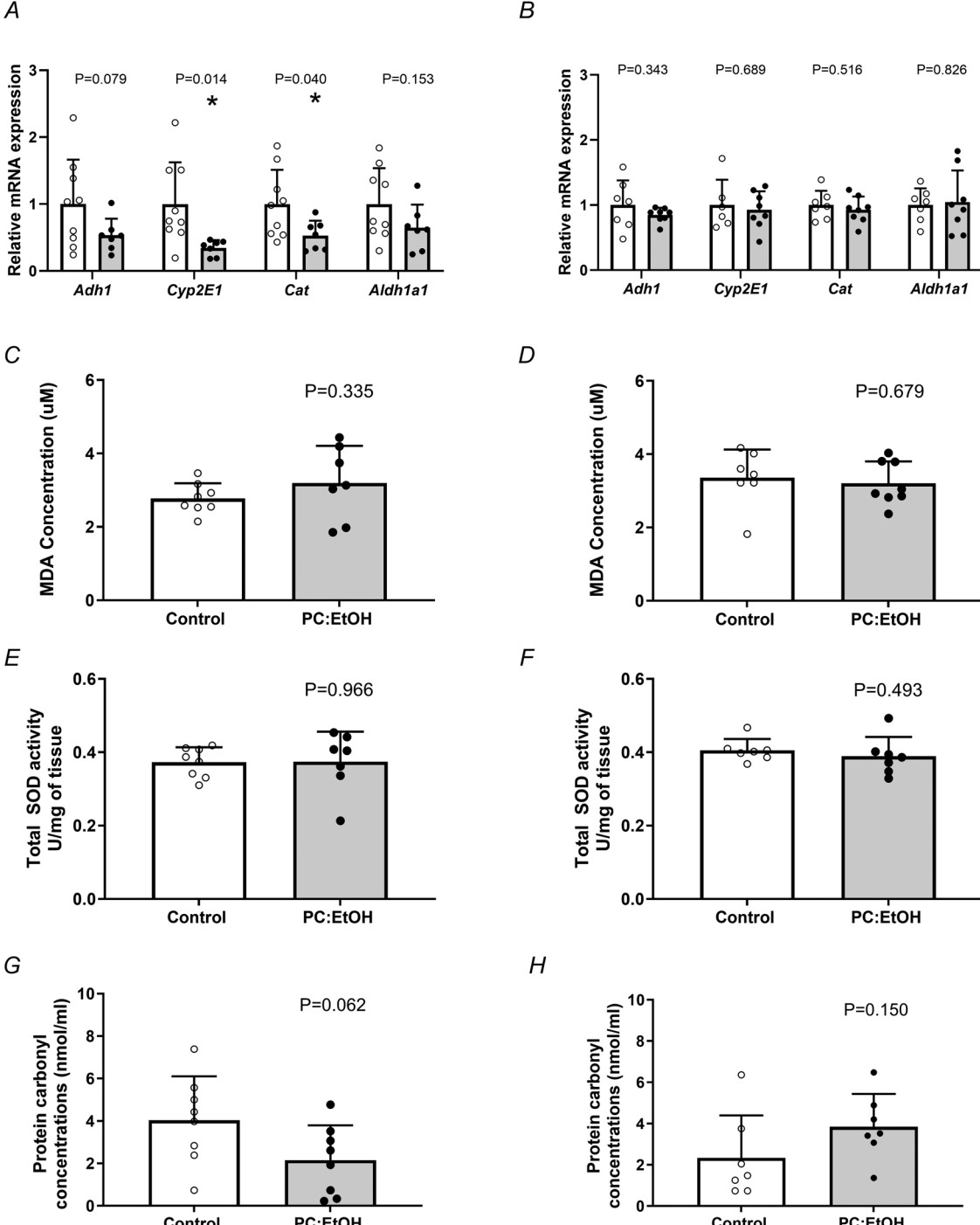

**Figure 2. The effects of periconceptional ethanol exposure (PC:EtOH) on hepatic mRNA expression of alcohol metabolism genes**

Alcohol dehydrogenase 1 (*Adh1*), cytochrome P450 2E1 (Cyp 2E1), catalase (Cat) and aldehyde dehydrogenase one family member A1 (Aldh1a1) at embryonic day (E) 5 (*A*) and E20 (*B*), concentrations of the MDA as assessed by TBARS assay at E5 (*C*) or E20 (*D*), SOD activity on E5 (*E*) and E20 (*F*) and protein carbonyl concentrations at E5 (*G*) and E20 (*H*) in the maternal liver. Data are presented as the mean + SD. Open circles (white bars) represent the control group and black circles (grey bars) the PC:EtOH exposed group. Nine controls and seven PC:EtOH samples were used for qPCR at E5. Seven controls and eight PC:EtOH samples were used for qPCR at E20 with one control sample removed for *Cyp2e1 m*RNA expression. Eight controls and seven PC:EtOH samples were used for TBARS

at E5, whereas seven controls and eight PC:EtOH samples were used for TBARS at E20. Eight controls and seven PC:EtOH samples were used for SOD at E5, whereas seven controls and seven PC:EtOH samples were used for SOD at E20. Eight controls and eight PC:EtOH samples were used for protein carbonyls at E5 and at E20 but one control and one PC:EtOH at E20 were not usable. $P < 0.05$ was considered statistically significant. **$P < 0.01$, *$P < 0.05$.

number of samples able to be analysed on a single western blot, the concentration of proteins that contribute to key complexes of the mitochondrial electron transport chain were then assessed separately for males and females. PC:EtOH did not affect protein expression of NDUFB8 (Complex I, $P = 0.426$), SDHB (Complex II, $P = 0.181$), UQCRC2 (Complex III, $P = 0.111$), MTCO1 (Complex IV, $P = 0.561$) or ATP5A (Complex V, $P = 0.214$) in the JZ of E20 placentas from male fetuses (Fig. 6*B* and *D*). Similarly, PC:EtOH did not affect protein expression of NDUFB8 (Complex I, $P = 0.731$), SDHB (Complex II, $P = 0.431$), UQCRC2 (Complex III, $P = 0.731$), MTCO1 (Complex IV, $P = 0.445$) or ATP5A (Complex V, $P = 0.232$) in the JZ of E20 placentas from female fetuses (Fig. 6*C* and *E*).

## Discussion

Alcohol exposure around the time of conception is common and, although many pregnant individuals cease drinking alcohol upon pregnancy recognition, early exposure may impair embryonic development and lead to deficits in the offspring. The mechanisms through which an early exposure alters development are probably the result of a combination of impacts on maternal physiology and direct actions of alcohol on embryonic cells present during exposure. The present study has demonstrated that alcohol exposure around conception has minimal effects on the mRNA expression of alcohol metabolism genes in the maternal liver and does not cause changes in markers of oxidative stress in the maternal liver either right at the end of alcohol exposure at E5 or 15 days later at E20. However, mtDNA content in the maternal liver was markedly reduced at E5 with levels still 40% lower at E20. This was associated with a significant reduction in the expression of key mitochondrial proteins at E20. This may lead to changes in liver function that could indirectly influence pregnancy outcomes. PC:EtOH also induced a small but significant reduction in mtDNA levels within the JZ of the placenta at E20, although no changes in mitochondrial protein levels were evident. There was also an upregulation of antioxidant enzymes within the JZ of PC:EtOH placentas and no signs of oxidative stress. These subtle changes in placental antioxidant status may be a compensatory adaptation to earlier changes in placental function that may have originated during the ethanol exposure. It is possible that alcohol caused a reduction in mtDNA in trophectoderm (TE) cells of the early blastocyst, similar to the reduction seen in the maternal

liver at that time point. Because these mitochondria regulate how the embryo and later child utilise energy substrates for growth and development, it is possible that the deficits in placental biology seen in late gestation are a consequence of mitochondrial dysfunction that was caused during the early alcohol exposure.

Our first major finding was that mtDNA content in the maternal liver was reduced by ~77% with mtDNA levels still 40% lower 15 days later at E20. Although a 77% reduction in mtDNA appears to be large, similar results have been reported previously in non-pregnant rodents. Mansouri et al. (1999) demonstrated that administering an intragastric binge dose of EtOH ($5 \text{ g kg}^{-1}$) to mice caused a 51% depletion of mtDNA only 2 h after treatment. They proposed that mtDNA depletion was not caused directly by the EtOH but rather by its metabolic products. It is suggested that hepatic mitochondrial loss is a result of inhibition of mitochondrial fusion and upregulation of mitochondrial membrane permeabilization after ethanol exposure (Naghdi et al., 2018). Interestingly, the large decrease in mtDNA seen in the current study at E5 did not translate into a reduction in all mitochondrial proteins measured at that time. Although expression of a key subunit of complex I was reduced, expression of SDHB, a component of complex II was increased. Interestingly, others have shown that, when complex I of the electron transport chain is inhibited, compensatory mechanisms can increase expression of complex II in an attempt to maintain energy metabolism (Ramanujan & Herman, 2007). This shift in complex protein expression may also contribute to changes in energy substrate usage and cellular signalling (Iverson et al., 2023), although additional studies would be required to investigate this further. Importantly, by E20, all mitochondrial proteins measured in the maternal liver were decreased. It is possible that the decrease in mitochondrial proteins is an adaptive mechanism to reduce ATP production and eliminate defective mitochondria to offset any increase in ROS production that may have occurred following alcohol metabolism.

Such changes in mitochondrial biology within the liver of the mother could, however, induce a range of consequences. A reduction in mitochondrial content may have resulted in changes to metabolism that could have indirectly affected the fetus and contributed to long-term disease in offspring. Placental and fetal development are strongly influenced by maternal glucose, amino acid and lipid availability. Mitochondrial dysfunction in

the maternal liver can result in impaired utilisation of energy substrates and decreased liver ATP production. The impact of such dysfunction on fetal outcomes has been studied in mice where disruption to mitochondrial proteins in the maternal liver can lead to altered lipid and glucose metabolism with direct consequences on the fetus (Bowman et al., 2019). Similarly, mouse studies have shown that, when liver dysfunction occurs for the first time during pregnancy, this can progress to more severe liver disease in the mother later in life (Hribar et al., 2024).

Future studies should investigate if PC:EtOH induced changes to liver mitochondria have changed the utilisation of energy substrates such as glucose and whether these changes might have contributed to poor fetal outcomes or deficits in the mother after pregnancy.

The data from the present study suggest that alcohol induced reductions in mitochondrial content in the maternal liver are not a consequence of oxidative stress caused by alcohol metabolism. We have not directly assessed ROS production during the alcohol exposure

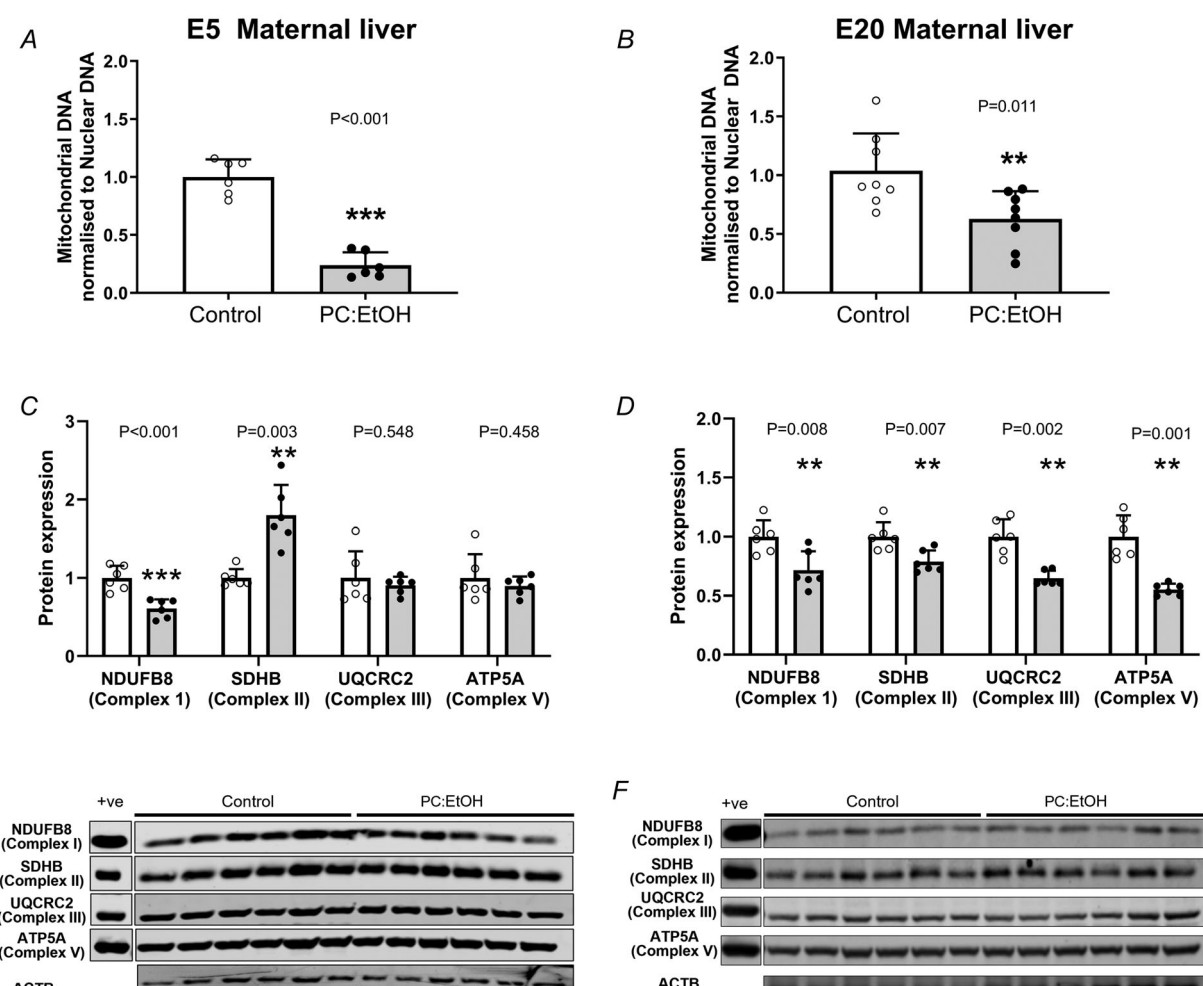

**Figure 3. Impact of periconceptional ethanol exposure (PC:EtOH) on mitochondrial content of the maternal liver**
Mitochondrial DNA (mtDNA) content at E5 (*A*) and E20 (*B*) and the expression of proteins that form complexes of the electron transport chain at E5 (*C*) and E20 (*D*) in the maternal liver. Representative western blot images from E5 shown in (*E*) and from E20 shown in (*F*). Mitochondrial DNA content was determined by normalising the qPCR amplification of mitochondrial DNA markers (*Mt-Dloop*, *Mt-DNA1* and *Mt-Nd4*) to nuclear DNA markers (*β-actin* and *Tet3*). Protein representative of complex I (NADH: ubiquinone oxidoreductase subunit B8 – NDUFB8), complex II (succinate dehydrogenase complex iron-sulphur subunit – SDHB), complex III (cytochrome b-c1 complex subunit 2 – UQCRC2) and ATP synthase (complex V, ATP synthase F1 subunit alpha – ATP5A). A positive control sample (+ve) consisting of isolated rat heart mitochondria was included on all blots. All protein analysis was normalised to expression of beta actin (ACTB). Data are presented as the mean + SD. Open circles (white bars) represent the control group and black circles (grey bars) the PC:EtOH exposed group. DNA was extracted from *n* = 6 per group for E5 and 8 per group at E20. Protein was extracted from *n* = 6 per group at E5 and E20. No data were excluded from any of these analyses. *$P < 0.05$ was considered statistically significant. **$P < 0.01$, ***$P < 0.001$.

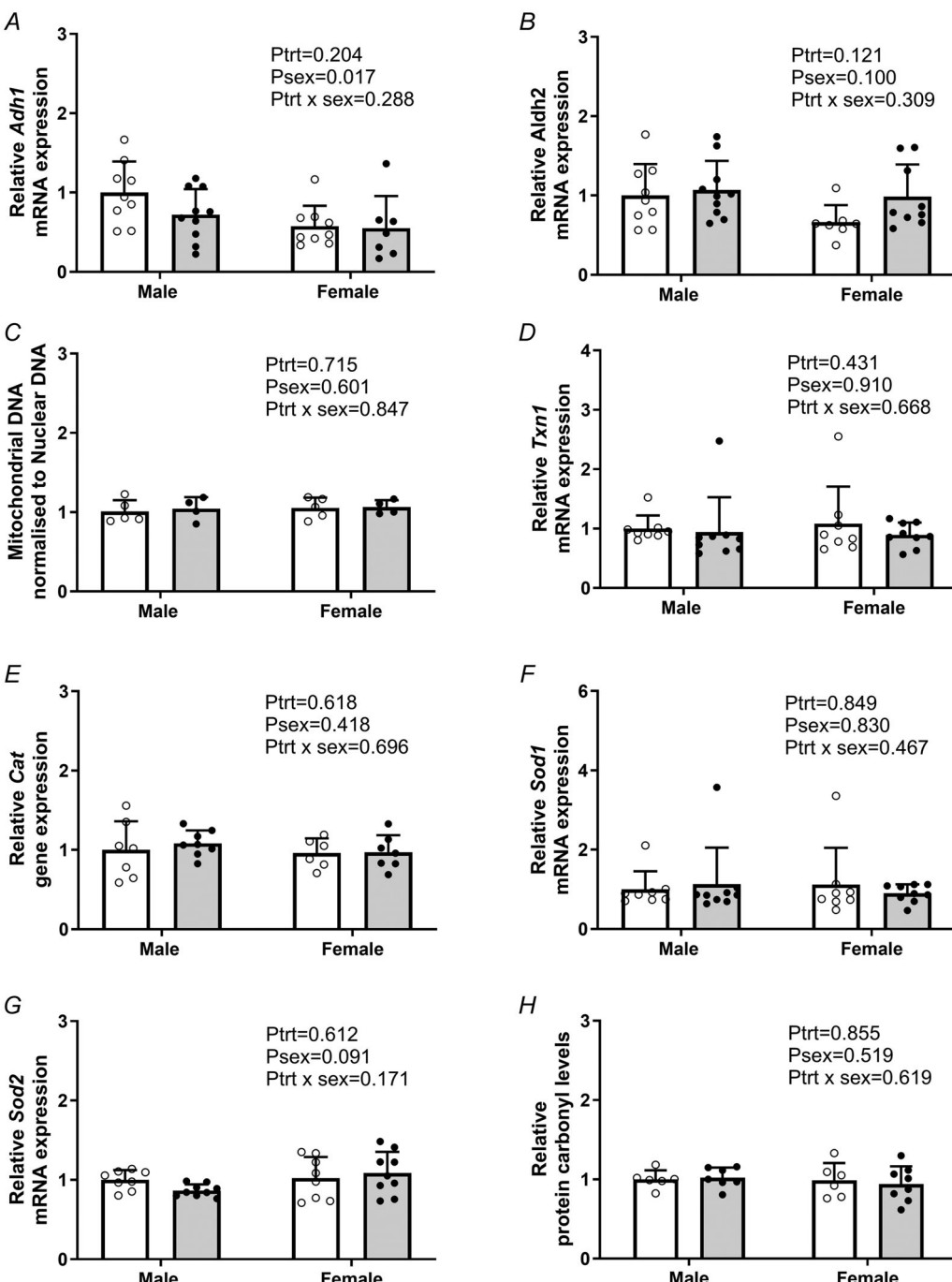

**Figure 4. Impact of periconceptional ethanol exposure (PC:EtOH) on alcohol metabolism genes and mitochondrial properties in the placental labyrinth**

mRNA expression of alcohol metabolism genes *Adh1* (alcohol dehydrogenase 1A) (*A*) and *Aldh2* (acetaldehyde dehydrogenase 2, mitochondrial) (*B*), mitochondrial DNA (mtDNA) content (*C*), mRNA expression of antioxidant enzymes *Txn1* (Thioredoxin 1) (*D*), *Cat* (catalase) (*E*), *Sod1* (superoxide dismutase 1) (*F*) and *Sod2* (superoxide dismutase 2) (*G*) and relative protein carbonyl levels (*H*) in the placental labyrinth. Data are presented as the mean + SD and analysed using a two-way ANOVA with treatment (control *vs*. PC:EtOH) and sex (male or female) as factors. If the ANOVA indicated an effect of treatment ($P_{trt} < 0.05$), sex ($P_{sex} < 0.05$) or an interaction between the two ($P_{trt} \times P_{sex} < 0.05$), then a Sidak multiple comparisons test was performed. Open circles (white bars) represent the control group and black circles (grey bars) the PC:EtOH exposed group. DNA was extracted from 4 to 5 LZ tissues per group. No DNA data were excluded from data analysis. RNA was extracted from 8–10 samples per group. Data were only excluded if a sample indicated failed amplification. Protein was extracted from $n = 6$–8 per treatment group for protein carbonyls.

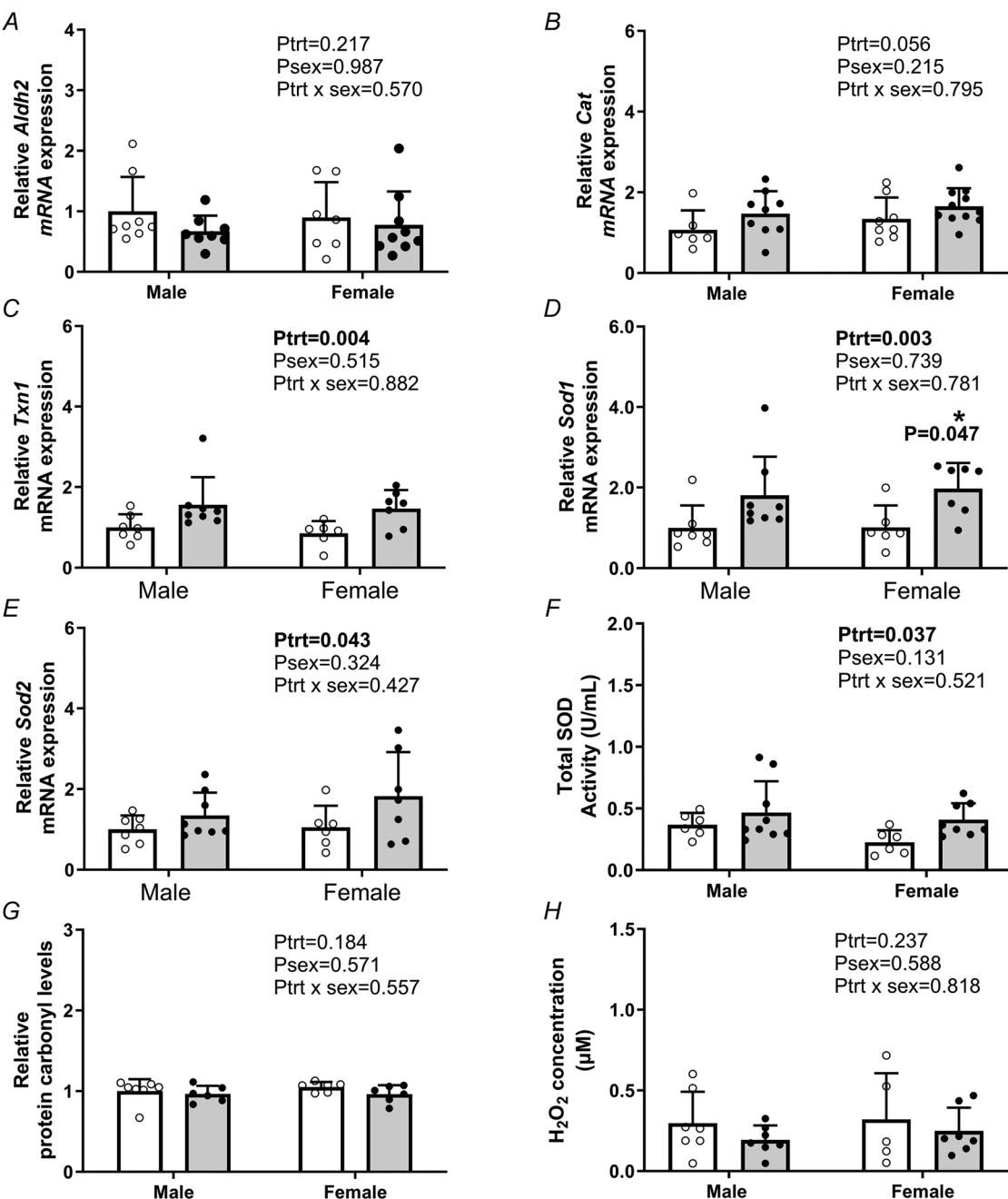

**Figure 5. Impact of periconceptional ethanol exposure (PC:EtOH) on alcohol metabolism genes and mitochondrial properties in the placental junctional zone**

mRNA expression of *Aldh2* (acetaldehyde dehydrogenase 2, mitochondrial) (*A*), *Cat* (catalase) (*B*), *Txn1* (thioredoxin 1) (*C*), *Sod1* (superoxide dismutase 1) (*D*) or *Sod2* (superoxide dismutase 2) (*E*), total SOD activity (*F*), relative protein carbonyl levels (*G*) or $H_2O_2$ concentrations (*H*) in placental JZ at E20. Data are presented as the mean + SD and analysed using a two-way ANOVA with treatment (control *vs.* PC:EtOH) and sex (male or female) as factors. If the ANOVA indicated an effect of treatment ($P_{trt} < 0.05$), sex ($P_{sex} < 0.05$) or an interaction between the two ($P_{trt} \times P_{sex} < 0.05$), then a Sidak multiple comparisons test was performed. If the Sidak test revealed a significant difference, this is indicated by an asterisk (*) when $P < 0.05$. Open circles (white bars) represent the control group and black circles (grey bars) the PC:EtOH exposed group. Six to nine samples were extracted per treatment group for qPCR/SOD protein activity/protein carbonyl and $n = 5$–7 samples were extracted for $H_2O_2$ concentrations. qPCR data were only excluded if a sample indicated failed amplification; no data were removed from the analysed SOD/protein carbonyls/$H_2O_2$ analyses.

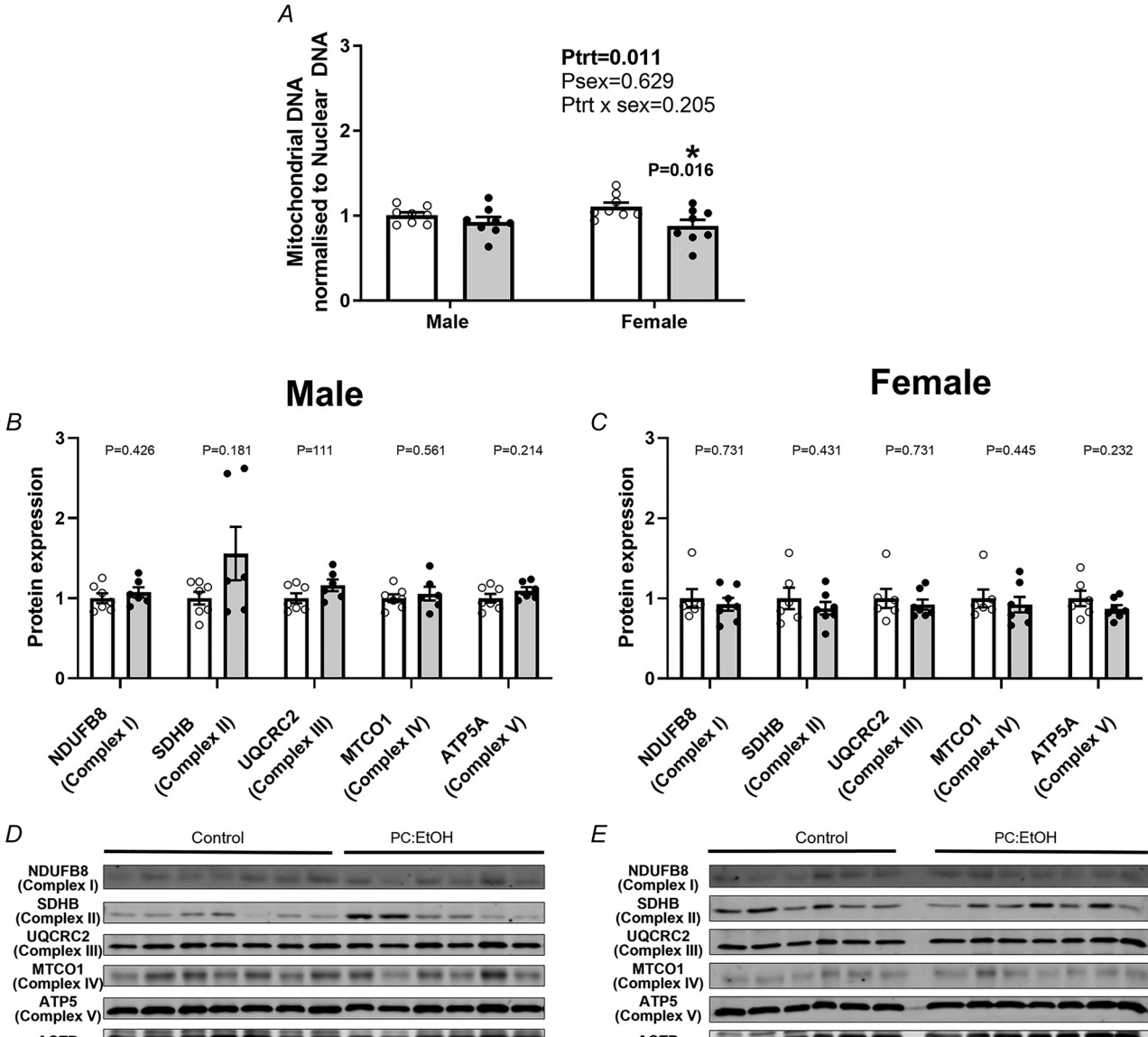

**Figure 6. Impact of periconceptional ethanol exposure (PC:EtOH) on mitochondrial content in the placental junctional zone**

Mitochondrial DNA (mtDNA) content in the placental JZ of males and females (*A*), and the expression of proteins that form complexes of the electron transport chain in males (*B*) and females (*C*). Representative western blot images from JZ tissue of male fetuses are shown in (*D*) and images from JZ tissue of female fetuses are shown in (*E*). Mitochondrial DNA content was determined by normalising the qPCR amplification of mitochondrial DNA markers (*Mt-Dloop*, *Mt-DNA1* and *Mt-Nd4*) to nuclear DNA markers (*β-actin* and *Tet3*). Protein representative of complex I (NADH: ubiquinone oxidoreductase subunit B8 – NDUFB8), complex II (succinate dehydrogenase complex iron–sulphur subunit – SDHB), complex III (cytochrome b-c1 complex subunit 2 – UQCRC2), complex IV (mitochondrially encoded cytochrome c oxidase I – MT-CO1) and ATP synthase (complex V, ATP synthase F1 subunit alpha – ATP5A). All protein analysis was normalised to expression of beta actin (ACTB). Data are presented as the mean ± SD. Open circles (white bars) represent the control group and black circles (grey bars) the PC:EtOH exposed group. DNA were extracted from eight samples per group. Protein was extracted from $n = 6$–7 per group for western blot analysis. All extracted samples were analysed with no data removed. *$P < 0.05$ was considered statistically significant.

and, instead, base this conclusion on markers of oxidative stress at E5 and E20. It is possible that excessive ROS were produced as a biproduct of alcohol metabolism during alcohol exposure from 4 days prior to conception and until 4 days after conception but that this had no long-term consequence on any of the markers of oxidative stress that we measured. It is possible that the reduction in mitochondrial content was an adaptive response to minimise ROS production and prevent oxidative stress. In contrast to the findings from our study, Lee et al. (2020) demonstrated that 2 weeks of ethanol consumption prior to pregnancy in mice increased the MDA levels within the liver of the dams at E15 and reduced mtDNA and also suggested that the damaging effects of alcohol on the maternal liver were largely through the resulting oxidative stress (Lee et al., 2020). They demonstrated that inhibition of CYP2E1 prevented the increase in MDA and reduction in mtDNA levels, prevented alcohol induced steatosis and improved overall liver health. This in turn ameliorated alcohol induced reductions in litter size. Because CYP2E1 mediated metabolism of alcohol is a major regulator of the ROS induced by alcohol, it was interesting to note that *Cyp2e1* and *Cat* were the only alcohol metabolising genes found to be reduced in the maternal liver in the present study at E5. Perhaps, a compensatory reduction in *Cyp2e1* at this age ameliorated further damage to liver health. However, this may have impaired alcohol metabolism, allowing a higher level of alcohol to reach the early embryo, if that alcohol consumption had continued. Interestingly, no changes in alcohol metabolising genes were identified at E20 either in the maternal liver or the placenta, suggesting that earlier exposure has not caused a long-term disruption of the mRNA expression of these enzymes. It remains unclear why the results from our study differ from those obtained by Lee et al. (2020) who used a lower dose of alcohol for a longer period of time. Such differences may be attributed to differences in study design or the fact that mice were used rather than rats.

Another finding of the present study was that mtDNA levels within the JZ of the placentas were also decreased with PC:EtOH reducing mtDNA by 15% in placentas from female fetuses compared to female controls at E20. Given that the degree of mtDNA loss at E20 in the maternal liver was just a fraction of the loss seen at E5, it is easy to speculate that, had we collected placental samples earlier in pregnancy, we might have seen a more overt reduction in mtDNA content. By contrast, mitochondrial protein levels in the JZ were not significantly affected by PC:EtOH. It is possible that mitochondrial protein levels had been decreased earlier but that compensatory repair mechanisms had restored protein levels by E20. It is also possible that the mitochondria that remain, are morphologically different such that DNA levels are lower despite not impacting mitochondrial protein levels.

Studies in women have shown that prenatal exposures to adverse factors such as smoking (Bouhours-Nouet et al., 2005) and diabetes (Hastie & Lappas, 2014) can result in lower placental mitochondrial content. Similarly, diet induced obesity has been shown to reduce mitochondrial content in placentas from pigs (Hu et al., 2019). However, such studies have not characterised the mechanisms by which periconceptional exposures impacts placental mtDNA content later in pregnancy. Janssen et al. (2015) investigated the impact of airborne particulate matter in women on placental mtDNA content and found that, although exposure across the entirety of pregnancy or exposure during the third trimester alone significantly reduced mtDNA content, exposure during the first trimester did not affect term mtDNA (Janssen et al., 2015). As such, the present study may be one of the first to report reduced placental mtDNA content in placentas collected close to term when the adverse exposure occurred during the periconceptional period. It is possible that the PC:EtOH induced accumulation of glycogen that we previously reported (Gardebjer et al., 2014) may be a result of the reduced mitochondrial capacity to utilise glucose for oxidative phosphorylation and so the glucose has been stored as glycogen, although further studies are required to investigate this further.

It is important to note that the placenta did not exist as a distinct organ during any stage of the ethanol exposure. It is therefore possible that PC:EtOH may have a caused a reduction in mtDNA in the oocyte before pregnancy, or before cells of the early embryo or even TE cells of the blastocyst, and that these early deficits resulted in the subtle 15% reduction in mtDNA still seen in the JZ at E20. During oocyte maturation, mtDNA content increases significantly, making the time leading up to ovulation a time of mitochondrial vulnerability (Kirillova et al., 2021). During early embryo development, mitochondrial biogenesis and therefore mtDNA replication pauses before resuming again as the embryo starts to differentiate into the inner cell mass and the TE (May-Panloup et al., 2021). It is possible that this also creates a period where impairment of mtDNA may have long lasting effects on the developing embryo. We have previously reported that this model of PC:EtOH does not affect TE cell counts but that CDX2 immunofluorescence, an indicator of TE pluripotency, was reduced (Kalisch-Smith et al., 2017). Interestingly, loss of pluripotency is known to be linked to an increase in mitochondrial biogenesis (Facucho-Oliveira & St. John, 2009). We show that mtDNA content is lower at E20 despite evidence of lower pluripotency at E5, suggesting that the lower mtDNA level is probably a consequence of mtDNA damage or mitophagy rather than a change in mitochondrial biogenesis linked with a change in cell fate. We also showed in our previous study that 5-methylcytosine was increased at E5 and so

have previous evidence of PC:EtOH induced changes to DNA methylation within the cells that form the placenta (Kalisch-Smith et al., 2019). Placental tissues taken from this same model of PC:EtOH were also found to have a reduced *S*-adenosylmethionine:*S*-adenosylhomocysteine ratio (SAM:SAH) (Steane et al., 2022) and sex specific changes in global DNA methylation at E20 (Steane et al., 2021), confirming that alcohol early in pregnancy caused changes in DNA methylation in cells of the embryo that resulted in changes in DNA methylation in the placenta late in pregnancy. Acetaldehyde is known to inhibit methionine synthase activity that results in a reduction of SAM (Kenyon et al., 1998). Oxidative stress is known to similarly reduce SAM via homocysteine conversion to glutathione (Kerksick & Willoughby, 2005). Both of these processes could have contributed to the shift in DNA methylation seen in this model of PC:EtOH. It is possible that alcohol metabolites and oxidative stress caused both loss of mtDNA and dysregulated DNA methylation and that these factors combined contributed to the long-term changes in the placenta close to term. It would be of interest to assess mtDNA and mitochondrial activity in the oocytes and TE in this model of PC:EtOH.

Despite the 15% reduction in mtDNA at E20 in the JZ, neither marker of oxidative stress was increased. This may suggest that, if oxidative stress was the mechanism by which alcohol caused a long-term reduction in mtDNA at E20, compensatory mechanisms have restored the cell such that oxidative stress is no longer present. The fact that both mRNA expression of multiple antioxidant enzymes and SOD activity were increased at E20 supports the idea that compensatory adaptations have restored the balance between ROS and antioxidants within the JZ.

Intriguingly, there were no differences in mitochondrial content, antioxidant expression or ROS observed in the placental labyrinth despite the fact that we have previously reported that PC:EtOH causes a number of other changes in that region (Kalisch-Smith et al., 2017). Sferruzzi-Perri et al. (2019) have previously shown that mitochondria within the two zones of the rodent placenta perform quite differently over gestation (Sferruzzi-Perri et al., 2019), which may suggest that one zone is more vulnerable to mitochondrial perturbations than the other. Such differences in mitochondrial activity in these zones probably reflects their different functions. It is possible that changes to mitochondrial activity in the placental labyrinth would result in impaired capacity to produce the ATP required for transport of nutrients to the fetus and so greater adaptive processes in this tissue may be required to support the survival of the fetus. By contrast, the ATP generated in the JZ is probably more strongly linked to the production of hormones that are secreted into the maternal circulation and so may be less strongly tied to fetal survival. Previous studies have reported that the JZ is more vulnerable to cellular stress related dysfunction.

Yung et al. (2012). reported that there are more ER cisternae in the JZ compared to the LZ and that genetic manipulation of *Eif2s1*, an essential ER stress regulating protein, can induce ER stress in the JZ but not the LZ. Future studies should explore how alcohol may induce different forms of cellular stress within each zone of the placenta and impact mitochondrial survival and function.

### Limitations and alternate proposed mechanisms

The present study demonstrates that alcohol has caused a reduction in mtDNA in the maternal liver but we have not identified the mechanism by which this occurs. We found no evidence of oxidative damage using the selected markers. Another limitation of the present study is that, although we have suggested that the reduction in mitochondrial content at E20 in the placenta are a consequence of direct alcohol effects on the TE, we have not measured mitochondrial content in the embryo. It is also possible that the alcohol induced effects on the JZ mitochondria at E20 are a consequence of an altered metabolic state in the mother linked with the changes in liver mitochondrial function. As such, future studies should characterise how this model of PC:EtOH impacts glucose homeostasis and metabolic hormones concentrations. Another limitation of this study was that we were not able to measure the activity of alcohol metabolising enzymes in the maternal liver. We also only measured activity of one antioxidant in the liver and placenta. In future studies, such parameters should be assessed to confirm the findings from this study.

### Conclusions

Overall, the findings from the present study suggest that short-term exposure to alcohol around conception leads to changes to mitochondria within the maternal liver for the duration of pregnancy. Perhaps, more notably, PC:EtOH altered the mitochondria DNA content, antioxidant gene expression and SOD activity in the JZ of the placenta. Because we have previously shown that the JZ is morphologically altered in this model, it is possible that the mitochondrial related changes reported in this study may have contributed to that phenotype. Future studies are needed to explore the mechanisms that link alcohol around conception to changes in mitochondria at E20. The present study highlights the need for continued efforts to increase public awareness of the potentially harmful effects of alcohol consumption during pregnancy, even when this is isolated to a short period around conception. Our study provides additional evidence to support current guidelines that recommend avoidance of alcohol when planning, as well as during pregnancy.

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

## Additional information

### Data availability statement

Data are available from the corresponding author upon raesonable request.

### Competing interests

The authors declare that they have no competing interests.

### Author contributions

J.S.M.C. and K.M.M. were responsible for conceptualisation. J.K.-S., L.K.A., K.M.M. and J.S.M.C. were responsible for animal work. S.E.S., T.D., J.S.M.C. and D.T.M. were responsible for formal analysis. S.E.S. and K.M.M. were responsible for funding acquisition. S.E.S., T.D., J.S.M.C., L.K.A., D.T.M. and K.M.M. were responsible for methodology. L.K.A., K.M.M. and J.S.M.C. were responsible for supervision. T.D. and J.S.M.C. were responsible for writing the original draft. S.E.S., L.K.A., J.S.M.C. and K.M.M. were responsible for reviewing and editing.

All authors have read and agreed to the final version of the manuscript submitted for publication.

## Funding

This research was funded by National Health and Medical Research Council for support of SES (grant number APP1191217) and KMM (grant number APP1078164).

## Acknowledgements

We acknowledge the assistance of Dr Nykola Kent, Arree Fielding and Isabella Anderson for assistance generating animal samples. We acknowledge Jenny Kuo and Rose Norton for assistance with preparation of qPCR data for maternal alcohol metabolism genes.

Open access publishing facilitated by The University of Queensland, as part of the Wiley - The University of Queensland agreement via the Council of Australian University Librarians.

## Keywords

alcohol metabolism, DOHAD, mtDNA, superoxide dismutase

## Supporting information

Additional supporting information can be found online in the Supporting Information section at the end of the HTML view of the article. Supporting information files available:

**Peer Review History**

