## [Peer Review History · The Journal of Physiology]

Effects of periconceptional ethanol on mitochondrial content and oxidative stress in maternal liver and placentas from male and female fetuses in rats

Sarah E Steane, Tulika Das, Jacinta I Kalisch-Smith, Dinithi T Mahaliyanage, Lisa K Akison, Karen M Moritz, and James S M Cuffe

DOI: 10.1113/JP287566

Corresponding author(s): James Cuffe (j.cuffe1@uq.edu.au)

The following individual(s) involved in review of this submission have agreed to reveal their identity: Hannah E.J. Yong (Referee #1)

Review Timeline:

Submission Date:	28-Aug-2024
Editorial Decision:	08-Oct-2024
Revision Received:	28-Nov-2024
Accepted:	23-Jan-2025

Senior Editor: Laura Bennet

Reviewing Editor: Janna Morrison

Transaction Report:

Dear Dr Cuffe,

Re: JP-RP-2024-287566 "Periconceptual ethanol exposure in rats reduces mitochondrial content in the maternal liver and the placenta" by Sarah E Steane, Tulika Das, Jacinta I Kalisch-Smith, Dinithi T Mahaliyanage, Lisa K Akison, Karen M Moritz, and James S M Cuffe

Thank you for submitting your manuscript to The Journal of Physiology. It has been assessed by a Reviewing Editor and by 2 expert referees and we are pleased to tell you that it is acceptable for publication following satisfactory revision.

REVISION CHECKLIST:

Please upload two versions of your manuscript text: one with all relevant changes highlighted and one clean version with no changes tracked. The manuscript file should include all tables and figure legends, but each figure/graph should be uploaded as separate, high-resolution files. The journal is now integrated with Wiley's Image Checking service. For further details, see: <https://www.wiley.com/en-us/network/publishing/research-publishing/trending-stories/upholding-image-integrity-wileys->

image-screening-service

We look forward to receiving your revised submission.

Yours sincerely,

Laura Bennet
Senior Editor
The Journal of Physiology

EDITOR COMMENTS

Reviewing Editor:

Comments to the Author:

This is a comprehensive investigation of the impact of periconceptional alcohol consumption on mitochondrial function in the placenta and liver.

Have the authors considered measuring ROS directly?

Should the title include the fact that sex is compared?

Line 119 - Please use 'humanely killed' rather than culled?

Did you use 1 placental per sex per dam or did you nest for dam?

Figure 1 - Lovely. Please indicate the number of dams used at each age (E5 and E20)? Please use 'humanely' killed.

Line 215 and 220 - Thank you for specifying sample size; however, this is not a sentence.

Please use SD rather than SEM as per journal guidelines.

Results - Please include the actual P value to 3 decimal places for all comparisons included in the text. Please add p values to figures and tables.

Did you test for outliers using Grubbs?

Please explain why samples were excluded in figure legends.

Line 393 and 414 and 466 and 538 - 'which' should be 'that'.

Line 414 - females

Line 435 - remove 'marked'.

Line 438 - remove 'impactful'

Lines 477-81 - Please edit for clarity.

Line 485 - 'overall' - do you mean 'poor' health? It isn't clear as written.

Line 507 - remove 'how' - suggest 'the mechanisms by which'

Line 510 - .. '.. alone significantly reduced mtDNA...'

Line 533 - which should be preceded by a comma.

Line 540 - the formatting of the ref is incorrect.

REFeree COMMENTS

Referee #1:

This is a neat and straightforward study by Steane et al. The authors showed that periconceptional ethanol can affect the maternal liver mitochondria and the placental junctional zone in a rat model. This study adds another insight into how periconceptional ethanol affects maternal and placental physiology at the level of the mitochondria. There are just several comments that require addressing.

Line 23-24: Please clarify if you are referring to liver or placenta in the first part of the point.

Figure 6: Why was a two-way ANOVA not performed for the mitochondrial DNA content as was done for Figures 4 and 5?

How relevant is the dosing of 12.5% v/v of alcohol to the typical periconceptional alcohol exposure in humans? Is this considered light, moderate or heavy drinking? The exposure is simply described as modest (line 558) in the conclusion, but the dose is more than double the 5% given in a mouse model by Lee et al. 2020. Please add more context to the discussion.

Please keep the nomenclature for the complexes consistent throughout the paper as it switches back and forth between Arabic numerals and Roman numerals (eg. lines 252 to 256, 459 to 460, Figure 3 legend etc.).

Referee #2:

The present study aimed to investigate the effects of alcohol consumption during the pre- and peri-conceptual period on maternal and placental metabolism. The methods were appropriate, and the study has been carried out extremely well. The manuscript is extremely well-written and a pleasure to read. The results are novel and suggest early alcohol exposure can have maternal and fetal consequences. I only have minor comments that I wish the authors to consider:

Minor

Line 23-24 - in what tissue?

Line 53: suggest including some indication of alcohol intake (in units) in these 50% of women (average, and extreme cases).

Line 94: This is probably my own ignorance on embryonic development and/or ethanol metabolism, but how would ethanol or acetaldehyde come into contact with the blastocyst and developing embryo before the maternal/fetal blood circulation has been established? If there is no contact, then how could there be a "direct effect" on mitochondria?

Line 114 - I appreciate that this is a well-established model, but does continuously drinking ethanol really recapitulate average or severe alcohol consumption in women trying to conceive (e.g. binge drinking)? Perhaps some justification of the model here would be useful for the non-expert. Have ethanol levels in the maternal circulation been measured in this model?

Line 201: Stated that the absolute concentration of H2O2 could not be measured accurately. But it was reported on line 242

Line 413: There seems to be something missing here (at least on my copy), the sentence is incomplete. Maybe the figure title?

Lines 465-472: This section is a bit vague. Suggest being a bit more specific about the metabolic consequences for ATP generation, but also ROS production. Is it possible that the reduction in mito content and protein expression of ETC is actually adaptive, in order to reduce ROS production and avoid oxidative stress?

Line 474-479: see comment above. Perhaps there was oxidative stress, and this led to an adaptive reduction in mito content, and then ROS levels returned to normal.

Line 514: see earlier comment. How could the egg and/or embryo come into contact with alcohol? Are you proposing epigenetic effects? If so, how, and what? Would be good to expand here.

END OF COMMENTS

Response to referees

Reviewing Editor:

1. This is a comprehensive investigation of the impact of periconceptional alcohol consumption on mitochondrial function in the placenta and liver.

We thank the editor for this comment

2. Have the authors considered measuring ROS directly?

Most reactive oxygen species are short lived and not stable enough to measure in frozen samples. We have measured H₂O₂ generation following the addition of SOD to tissue lysate, but this is still an indirect measure. To directly measure ROS, a separate set of animal would have been required. As such, this was not feasible in this study.

3. Should the title include the fact that sex is compared?

We have reworded the title and included the fact that sex was assessed while keeping within the 150 character limit.

4. Line 119 - Please use 'humanely killed' rather than culled?

We have made this correction

5. Did you use 1 placental per sex per dam or did you nest for dam?

Yes, only one placenta per sex per dam was used for each experiment. We have stated this in line 134

6. Figure 1 - Lovely. Please indicate the number of dams used at each age (E5 and E20)? Please use 'humanely' killed.

We have now updated the figure and included how many animals killed at each age per group and used the term humanely killed. We also updated the figure to try and make it visually easier to see for people who are colour blind and made the figure using Biorender as per the Journal of physiology guidelines.

7. Line 215 and 220 - Thank you for specifying sample size; however, this is not a sentence.

Thank you for highlighting this mistake. We have now fixed this.

8. Please use SD rather than SEM as per journal guidelines.

We have now changed this to SD on all our figures

9. Results - Please include the actual P value to 3 decimal places for all comparisons included in the text. Please add p values to figures and tables.

We have updated the P values to make sure they are included in text and on figures.

10. Did you test for outliers using Grubbs?

No outliers were removed based on any statistical approach, we feel it most appropriate to include all data when possible. The number of data points presented for various sample analyses is dependant on the number of samples extracted for each process. Given the limited tissue size and the multiple extraction methods required, we have not used all samples for all analyses. For western blotting, n= 6-7 tissues were extracted per group and all samples were included in the dataset. For all other analyses, one or more samples was occasionally lost during the process of sample preparation (RNA extraction or cDNA synthesis steps, sample preparation for tissue enzyme activity or ROS marker concentration). Similarly, occasionally a sample did not amplify during the qPCR analysis. The sample sizes used for each protocol are now included in the figure legends.

11. Please explain why samples were excluded in figure legends.

We have now included why samples were excluded in figure legends (as per the comment above).

12. Line 393 and 414 and 466 and 538 - 'which' should be 'that'.

This has been corrected

13. Line 414 – females

This has been corrected

14. Line 435 - remove 'marked'.

This has been corrected

15. Line 438 - remove 'impactful'

This has been corrected

16. Lines 477-81 - Please edit for clarity.

We have now edited this for clarity

17. Line 485 - 'overall' - do you mean 'poor' health? It isn't clear as written.

We have now edited this for clarity

18. Line 507 - remove 'how' - suggest 'the mechanisms by which'

Thank you for this suggestion. We have now edited this as suggested

19. Line 510 - .. '!.. alone significantly reduced mtDNA...'

Thank you for this suggestion. We have now edited this as suggested

20. Line 533 - which should be preceded by a comma.

Thank you for this suggestion. We have now edited this as suggested

21. Line 540 - the formatting of the ref is incorrect.

Thank you for this suggestion. We have now edited this as suggested

REFEREE COMMENTS

Referee #1:

1. This is a neat and straightforward study by Steane et al. The authors showed that periconceptional ethanol can affect the maternal liver mitochondria and the placental junctional zone in a rat model. This study adds another insight into how periconceptional ethanol affects maternal and placental physiology at the level of the mitochondria. There are just several comments that require addressing.

We thank Reviewer 1 for this feedback.

2. Line 23-24: Please clarify if you are referring to liver or placenta in the first part of the point.

We have now edited this to make sure it is clear we are referring to the liver.

3. Figure 6: Why was a two-way ANOVA not performed for the mitochondrial DNA content as was done for Figures 4 and 5?

We agree that this should have been completed as a two-way ANOVA and we have now edited the manuscript and changed the relevant text to reflect this change.

4. How relevant is the dosing of 12.5% v/v of alcohol to the typical periconceptional alcohol exposure in humans? Is this considered light, moderate or heavy drinking? The exposure is simply described as modest (line 558) in the conclusion, but the dose is more than double the 5% given in a mouse model by Lee et al. 2020. Please add more context to the discussion.

The best way to compare how different forms of alcohol exposure impact outcomes in rodents is to measure blood alcohol content (BAC). We have measured BAC levels attained in this model previously and it peaks at ~0.18% 30 minutes after being offered the diet. However, it quickly drops down and reaches levels of about 0.07% and 0.05% at 3h and 5h respectively (18). So each day, there is likely a short period of time when the levels are relatively high but for the majority of time, the levels are low. For comparison, a 55 kg woman reaches a BAC of 0.14-0.16% following consumption of 4-5 standard drinks over 2-3 hours and it takes 6-7 hours for the alcohol to be eliminated from her body. Our model therefore models this type of drinking behaviour occurring daily over a 8 day period and

would be regarded as heavy alcohol use (see explanation in response to reviewer 2, comments 2 and 4).

The study by Lee et al, has not measured BAC (not measured in any papers published by that group) so it is difficult to directly compare. Other studies in the field use much higher doses with our work somewhere in the middle. We have added information about the BAC reached in this model to the methods section of the paper and commented on differences in study design between mice and rats. We have also made sure to remove the word modest from the discussion.

5. Please keep the nomenclature for the complexes consistent throughout the paper as it switches back and forth between Arabic numerals and Roman numerals (eg. lines 252 to 256, 459 to 460, Figure 3 legend etc.).

We have now gone through the manuscript and corrected throughout.

Referee #2:

The present study aimed to investigate the effects of alcohol consumption during the pre- and peri-conceptual period on maternal and placental metabolism. The methods were appropriate, and the study has been carried out extremely well. The manuscript is extremely well-written and a pleasure to read. The results are novel and suggest early alcohol exposure can have maternal and fetal consequences. I only have minor comments that I wish the authors to consider:

Minor

1. Line 23-24 - in what tissue?

We have now edited this to make sure it is clear we are referring to the liver.

2. Line 53: suggest including some indication of alcohol intake (in units) in these 50% of women (average, and extreme cases).

The data on prevalence of alcohol consumption is much more reliable than the data on number of units of alcohol intake. Often studies that investigate the amount consumed obtain this information retrospectively by survey and participants underestimate how many days per week they drink alcohol and how much they drink per day. Data obtained from these studies is then usually groups based on units of alcohol and prevalence. An example of how this grouping is done based on the McCormack study cited in the manuscript is "low" (less than 7 drinks per week up to 2 drinks at a time, moderate, (less than 7 drinks per week up to 4 drinks at a time, binge (less than 7 drinks per week but more than 4 drinks at a time) or heavy (more than 7 drinks per week). It is likely that within the heavy category there are a number of individuals that are much higher than the 7 standard drinks and they may binge multiple times per week. The McCormick study cited on line 53 did report on alcohol intake in the 50% of women who drank between conception and pregnancy recognition and the most common pattern of alcohol exposure was heavy alcohol use. We have included a

statement indicating that heavy drinking is common in women of reproductive age after we cite the McCormack study.

3. Line 94: This is probably my own ignorance on embryonic development and/or ethanol metabolism, but how would ethanol or acetaldehyde come into contact with the blastocyst and developing embryo before the maternal/fetal blood circulation has been established? If there is no contact, then how could there be a "direct effect" on mitochondria?

Prior to establishment of the maternal fetal interface at the placenta, the early embryo relies on oviductal and uterine fluid secretions for nourishment. Studies conducted in the early 1980's demonstrated that following alcohol exposure in rats, the oviductal and uterine secretions contained ethanol although they were lower than plasma levels. One study that investigated ethanol and acetaldehyde levels in rats after alcohol consumption demonstrated similar concentrations of ethanol in plasma and uterine tissue following alcohol consumption. That same study actually showed that acetaldehyde levels were higher in the uterus than in plasma. We had previously stated in the introduction that alcohol is permeable through cell membranes and that the embryo and fetus is exposed. We have now added additional details at line 64 and 98.

4. Line 114 - I appreciate that this is a well-established model, but does continuously drinking ethanol really recapitulate average or severe alcohol consumption in women trying to conceive (e.g. binge drinking)? Perhaps some justification of the model here would be useful for the non-expert. Have ethanol levels in the maternal circulation been measured in this model?

Yes, alcohol levels have been measured in this model previously. See response 4 to reviewer 1 (peaks at ~0.18% BAC at 30 minutes after being offered the diet, quickly drops to 0.07% and 0.05% at 3h and 5h respectively (18). So, each day, there is likely short period of time when the levels are quite high but for the majority of time, the levels are relatively low. For comparison, a 60 kg woman reaches a BAC of 0.12-0.16% following consumption of 4-5 standard drinks over 2-3 hours and it takes 6-7 hours for the alcohol to be eliminated from her body. We have now made it clear in the manuscript what the BAC is for this model.

This BAC equates to women drinking half a bottle of wine in an evening for 8 days in a row around the time of conception. This would fall into the heavy drinking category which is the most common type of alcohol consumption during the periconceptional period (see response to comment 2). We do acknowledge that this would be quite high for anyone planning a pregnancy, but more than 50% of pregnancies are unplanned. Furthermore, this high level of drinking is considered at the increasing risk category (risk 5 on a scale of 1-6) if it occurs every week of the year according the UK typical weekly drinking scale. But there are likely many additional women who drink moderate levels of alcohol but then drink high levels for a week or so only at certain times of the year, such as during holiday seasons with one study showing that high drinking 25-34 year olds might drink an additional 18 units of alcohol per week during holiday season(Bellis et al., 2015).

5. Line 201: Stated that the absolute concentration of H₂O₂ could not be measured accurately. But it was reported on line 242

H₂O₂ production in response to addition of SOD was assessed using an H₂O₂ assay throughout. This is not the native H₂O₂ present in the tissue but the amount of H₂O₂ able to be produced by the tissue when the required enzyme was added. I have altered the statement on what was line 242 to make this clear.

6. Line 413: There seems to be something missing here (at least on my copy), the sentence is incomplete. Maybe the figure title?

The formatting of the PDF did push the title of the figure off the page. We have made sure that this has been corrected.

7. Lines 465-472: This section is a bit vague. Suggest being a bit more specific about the metabolic consequences for ATP generation, but also ROS production. Is it possible that the reduction in mito content and protein expression of ETC is actually adaptive, in order to reduce ROS production and avoid oxidative stress?

Thank you for this suggestion. We have highlighted the possibility of this being an adaptive response to prevent oxidative stress and now elaborated on the metabolic consequences that may be possible.

8. Line 474-479: see comment above. Perhaps there was oxidative stress, and this led to an adaptive reduction in mito content, and then ROS levels returned to normal.

We have made sure to highlight this possibility in this section also.

9. Line 514: see earlier comment. How could the egg and/or embryo come into contact with alcohol? Are you proposing epigenetic effects? If so, how, and what? Would be good to expand here.

Both the egg and the embryo are known to be exposed to relatively high levels of alcohol following alcohol consumption (see response to comment above). We do expect that the effects of alcohol on development and long-term outcomes are mediated at least in part by epigenetic effects. We have published some data on epigenetic changes that occur in this model previously and we have expanded this section to detail how we think these ideas link together.

Bellis MA, Hughes K, Jones L, Morleo M, Nicholls J, McCoy E, Webster J & Sumnall H. (2015). Holidays, celebrations, and commiserations: measuring drinking during feasting and fasting to improve national and individual estimates of alcohol consumption. *BMC Med* **13**, 113.

Dear Dr Cuffe,

Re: JP-RP-2024-287566R1 "Effects of periconceptional ethanol on mitochondrial content and oxidative stress in maternal liver and placentas from male and female fetuses in rats" by Sarah E Steane, Tulika Das, Jacinta I Kalisch-Smith, Dinithi T Mahaliyanage, Lisa K Akison, Karen M Moritz, and James S M Cuffe

We are pleased to tell you that your paper has been accepted for publication in The Journal of Physiology.

Yours sincerely,

Laura Bennet
Senior Editor
The Journal of Physiology

If you would like to receive our 'Research Roundup', a monthly newsletter highlighting the cutting-edge research published in The Physiological Society's family of journals (The Journal of Physiology, Experimental Physiology, Physiological Reports, The Journal of Nutritional Physiology and The Journal of Precision Medicine: Health and Disease), please click this link, fill in your name and email address and select 'Research Roundup':
<https://www.physoc.org/journals-and-media/membernews>

- You can help your research get the attention it deserves! Check out Wiley's free Promotion Guide for best-practice recommendations for promoting your work at: www.wileyauthors.com/eoo/guide. You can learn more about Wiley Editing Services which offers professional video, design, and writing services to create shareable video abstracts, infographics, conference posters, lay summaries, and research news stories for your research at: www.wileyauthors.com/eoo/promotion.

EDITOR COMMENTS

Reviewing Editor:

Comments to the Author:

Thank you for addressing the reviewers comments.

REFEREE COMMENTS

Referee #1:

The authors have made changes as requested by the editor and reviewers and provided additional information as needed to support their points. I am satisfied with the current version and support the acceptance of the manuscript.

Referee #2:

The authors have addressed all my concerns.